# An Analytical Model for Describing the Power Coupling Ratio between Multimode Fibers with Transverse Displacement and Angular Misalignment in an Optical Fiber Bend Sensor

**DOI:** 10.3390/s19224968

**Published:** 2019-11-14

**Authors:** Wern Kam, Yong Sheng Ong, Sinead O’Keeffe, Waleed S. Mohammed, Elfed Lewis

**Affiliations:** 1Optical Fibre Sensors Research Centre (OFSRC), Dept. of Electronic and Computer Engineering, University of Limerick, V94 T9PX Limerick, Ireland; oys89n0e@hotmail.com (Y.S.O.); Sinead.OKeeffe@ul.ie (S.O.); elfed.lewis@ul.ie (E.L.); 2Center of Research in Optoelectronics, Communication and Computational Systems (BU-CROCCS), School of Engineering, Bangkok University, Pathumthani 12120, Thailand; wsoliman@gmail.com

**Keywords:** optical coupling efficiency, axial displacement, angular misalignment, fiber optic sensor, geometrical optics

## Abstract

The power coupling ratio between step-index multimode fibers caused by combined transversal and angular misalignment is calculated. A theoretical description of the coupling efficiency between two optical fibers based on geometrical optics is provided. The theoretical calculations are collaborated by experiments, determining the power coupling ratio between three output fibers with an axial offset and angular misalignment with a single input fiber. The calculation results are in good agreement with experimental results obtained using a previously fabricated optical fiber sensor for monitoring physiological parameters in clinical environments. The theoretical results are particularly beneficial for optimizing the design of optical fiber bending sensors that are based on power coupling loss (intensity) as the measurement interrogation requires either axial displacement, angular misalignment, or both.

## 1. Introduction

Optical fiber misalignment often occurs when joining two fibers in a permanent splice or using connectors. Transmission loss between connecting fibers can be due to intrinsic losses that are caused by a parameter mismatch between the fibers e.g., core diameter, numerical aperture (NA), different types (graded index or step-index). Extrinsic losses are incurred by geometrical misalignment of the fiber ends such as the presence of an air gap between the end faces [1,2], fiber tilt misalignment [3] or radial offset [4]. Provided the attenuation is limited at the plastic optical fiber (POF) connection, accurate estimation of the power coupling transmission between fibers can be achieved.

In some optical fiber sensors, geometrical misalignment between the fibers has been utilized as a basic configuration to modulate the light transmission for parameter detection [5]. These sensors measured the parameter by allocating a fiber displacement in lateral, longitudinal, angular, or differential deviations and analyzing the changed of transmission loss [6]. Sensors utilizing this measurement approach have also been implemented for the measurement of vibration [7], temperature [8], pressure [9], displacement [10,11], and medical bending monitoring [12,13]. In all such cases, an accurate analytical calculation of the transmission power (loss) between the fibers is essential to evaluate the measurement parameters such as the range before developing the sensor. 

For the case of the fiber based bending sensor of this investigation, multiple approaches have been previously developed using fiber optics. One approach involves the use of fiber Bragg grating (FBG) as the bending sensor where the external bending changes the pitch of the grating and thereby modulates the wavelength of the reflected light [14,15]. Another example of a bending sensor for monitoring seated spine posture was developed by Dunne et al. [16]. A side abraded POF was used to allow light leakage from the polished region in order to detect spine motion. The same sensing technique using a side-polished optical fiber was also applied to design a knee sagittal monitoring sensor [17]. Sensors using such designs have incurred the limitations of mechanical strength and bending at large angles or overstain, all of which result in permanent damage to the fiber sensor. Zawawi et al. [18] also developed an intensity modulated fiber sensor for spine monitoring using an aluminum foil placed at the end tips between the input and output fiber to allow part of the source light to be reflected into reference fiber and partly transmitted to the output fiber. The sensor demonstrated accurate results but the long term stability of device was found to be inadequate for repeated use in a clinical setting.

A theoretical analysis of the power coupling loss due to angular misalignment has previously been presented by Opielka et al. [19]. Gao et al. presented a similar angular misalignment analysis but with different numerical apertures (NA) between the input and output optical fibers used in sensing applications [20]. The work described in this article presents a theoretical consideration of the geometrical optics to describe the power coupling between the multimode fibers when subjected to combined lateral and angular misalignment. In the second section, a POF sensor based on tilt angle loss between the source transmitting fiber and three axially misaligned receiving output fibers is introduced and experimental results are included. The development of the theoretical model to directly complement the experimental measurements obtained using a real sensor system as reported in this article is the first instance that such an approach has been reported in the case of a real clinical based sensor system [21]. The results of the theoretical analysis formulated in the first section based on the optical fiber sensor configuration are directly compared with the experimental results. The theoretical investigation is significant as it facilitates the simulation of the parameters required for sensor based on transmission loss prior to fabricating the sensor. The result of the theoretical predictions is also beneficial for accurately calculating the transmission loss or power coupling between fibers with lateral and angular misalignment at the fiber coupling joint and hence determining the required sensor configurations for a wide range of other potential applications including structural monitoring and pressure monitoring.

## 2. Definitions and Assumptions

The calculation of power coupling between the input and output optical fibers was made based on the graphical explanation illustrated in Figure 1. The figure shows two fibers where fiber 1 is the input fiber (source transmitting) and fiber 2 is the receiving fiber with transverse misalignment and tilted at an angle with respect to the transmission fiber. Both fibers meet at the plane S with a lateral misalignment displacement, d and tilted at an angle, α (later referred as bending angle for the POF sensor). The theoretical analysis was performed based on several assumptions. First, all fibers in the calculation and experiment are identical step-index multimode fibers with the same numerical aperture (NA) and core diameter. This analytical approach also assumed a uniform mode power distribution in the multimode fiber. The model also neglects the effect of leaky modes.

## 3. Calculations

The power coupling efficiency received by fiber 2 was attributed to lateral and angular misalignment and calculated by integrating the solid angle Ω and overlap NA area at the plane S. Figure 2 shows the overlapping light delivery and capture cones of both fibers where the shaded area is integrated using the defined parameters and geometry. Based on the principles of geometrical optics, the power coupling between multimode fibers attributed to lateral and angular misalignment can be derived step-by step as shown below. Generally, the power transmitted within fiber 2 may be written as follows [19]:(1)$$P_{2}=\int_{S}\int_{\Omega_{2}}L\;cos\theta \;d\Omega_{2}dS_{2}$$
where L is the radiance based on the power distribution in the plane S and *θ* is the radiating area element within the overlap area from fiber 1. For the case of uniform power distribution, *L* is constant in the overlapping region. Considering the inner integration of Equation (1), the power density, *P*_2_^′^ is evaluated as:(2)$$P_{2}\prime =\int_{\Omega_{2}}L\cos\theta \;d\Omega_{2}$$
where the definition of solid angle, Ω at the overlapping region is represented by:(3)$$\Omega =\frac{A_{o}}{R^{2}}$$
where *A_o_* is the spherical surface area and *R* is the radius of the considered sphere, (here referred to the length of the captured cone extended from the fiber) as shown in Figure 2. Referring to Figure 2b, the overlapping area of the solid angle (Ω), $A_{o}$ for the NA value of the fibers (0.51) is integrated for each equal quadrant as the parameters shown in Figure 2b:(4)$$\Omega_{o}=\frac{4}{R^{2}}\int_{\beta =0}^{\beta_{max}}\int_{\rho_{min}}^{\rho_{max}}dA_{o}$$
(5)$$dA_{o}=\;R^{2}\;sin\theta \;d\theta \;d\beta$$
where *β* and *ρ* are the angle and radius integration point limits of the one quadrant (shaded area) illustrated in Figure 2b, and *θ* is the angle between the extended radiating area and the cone axis of fiber F1. 

For the case of combined transverse offset and angular misalignment coupling, the limit of the inner integral within the overlap area of one quadrant (hatched area) as illustrated in the Figure 2b can be calculated as follows:(6)βmax=arccos(U2ρ)
(7)$$U_{y}=\sqrt{{\left(y+t\right)}^{2}+x^{2}}$$
(8)$$U_{x}=\sqrt{y^{2}+{\left(x+t\right)}^{2}}$$
where *x* is the transverse displacement in the x-axis for the input fiber F1, *y* is the displacement in the y-axis for the input fiber F1, and *t* is the displacement of the F2 cone on the plane S due to an angular misalignment (tilt) in the y-axis. Equations (7) and (8) are applied with the integral limit corresponding to the direction of angular misalignment in y-axis and x-axis, respectively. The other integral limits are as follows: (9)$$\rho_{min}=R\frac{\sin\frac{\gamma }{2}}{\cos\beta }$$
(10)$$\sin\gamma =\frac{U}{R}$$
(11)$$\rho_{max}=R\sin\theta_{c}$$
where *γ* corresponds to the angle between the extended radiation at the lateral and angular misalignment and the normal of the emitting surface as presented in Figure 2a. *θ_c_* is the maximum acceptance angle (critical angle) for both identical fibers and corresponds to the angle *θ* in Figure 2a. The integral limit β_max_ can be further simplified as follows: (12)βmax=arccos(sinγ2sinθc)

Substituting the integration limits from Equations (9) to (12) and applying these to the overlap region of the solid angle (as represented in Figure 2) into Equation (2), it is possible to express the power density as follows: (13)$$P_{2}\prime =4\int_{\beta =0}^{\beta_{max}}\int_{\rho_{min}}^{\rho_{max}}L\;cos\theta \;sin\theta \;d\theta \;d\beta$$

Substituting sin θ = ρ/R, the inner integration in power density equation can be replaced with the following: (14)sinθmin=sinγ2cosβ

(15)$${\sin\theta }_{max}=\sin\theta_{c}$$

A solution of Equation (13) using the integral limits defined above yields the following result: (16)$$P_{2}^{\prime }=2L\left(sin^{2}\theta_{c}\beta_{max\;}-\;sin^{2}\frac{\gamma }{2}tan\beta_{max}\right)$$

The transmitted power in fiber 2 is found by integrating over the area element at the endface of the input fiber as follows: (17)$$P_{2}=\int_{0}^{r_{max}}\int_{0}^{2\pi }P_{2}^{\prime }\;d\theta \;r\;dr.$$

The *r_max_* value represents the boundary radius where the acceptance cone of both fibers no longer overlap at the plane S due to lateral and angular misalignment. In the case of the solution for a step index fiber, the *r_max_* value is the radius of the multimode fiber, *a*. Hence, the power coupling in fiber 2 including transmission loss due to transverse and angular misalignment may be written as: (18)$$P_{2}=2\pi a^{2}L\;\left(NA^{2}\beta_{max\;}-\;sin^{2}\frac{\gamma }{2}tan\beta_{max}\right).$$

Simplifying Equation (12) into: (19)βmax=arctan(sin2θcsin2γ2−1)12
and substituting Equation (19) into Equation (18), the power transmitted into fiber 2 may be expressed as follows:(20)P2=2πa2L (NA2arctan(sin2θcsin2γ2−1)12− sin2γ2(sin2θcsin2γ2−1)12).

The coupling efficiency is calculated as the ratio between received power in fiber 2 over the total power in the transmitting fiber, and is thus defined as follows: (21)$$\eta =\frac{P_{2}}{P_{I}}$$

(22)$$\eta =\frac{2}{\pi NA^{2}}\;\left(NA^{2}\beta_{max\;}-\;sin^{2}\frac{\gamma }{2}tan\beta_{max}\right).$$

### 3.1. Solution for a Two Step-Index Multimode Fibers of Different Numerical Aperture Values

This section discusses the case where both fibers have different Numerical Aperture (NA) values and where NA_1_ < NA_2_ and NA_1_ and NA_2_ represent the numerical aperture of fibers 1 and 2, respectively. Figure 3 shows the overlapping of light delivery and capture cones of two fibers with different NA values when undergoing both lateral and angular misalignment. In this case, the quadrant of the overlapping cone (shaded area) is not equal, as illustrated in Figure 3. It was necessary for the integration of the overlapping cones to be performed separately for quadrant areas 1 and 2. The integrated total power P_2_ of fiber 2 is thus the sum of P_a1_ and P_a2_, where P_a1_ is the transmitted power within overlap region 1 and P_a2_ is the power within overlap region 2.

In this case, it is also necessary that the integration limits, β and ρ be evaluated separately for different overlapping quadrant areas by first evaluating the angles ε_1_ and ε_2_ as illustrated in Figure 3a. The angle ε_1_ represents the angle between the cone axis of transmitting fiber to the center axis of intersection plane between the overlapping cone (hatched area) and ε_2_ is the angle between the cone axes of receiving fiber to the intersection plane between the overlapping cones. These angles are related to the angle between the extended radiation at the lateral and angular misalignment and the normal of the emitting surface, γ originally defined in Equation (10) as follows: (23)$$\varepsilon_{1}=\arctan\left(\frac{co{s\theta }_{2c}}{co{s\theta }_{1c}\;sin\gamma }-\frac{1}{tan\gamma }\right)$$
(24)$$\varepsilon_{2}=\arctan\left(\frac{co{s\theta }_{1c}}{co{s\theta }_{2c}\;sin\gamma }-\frac{1}{tan\gamma }\right)$$
where $\theta_{1c}$ and $\theta_{2c}$ is the acceptance angle of fiber 1 and fiber 2, respectively. The value of γ is the sum of both ε_1_ and ε_2_ as illustrated in Figure 3a. The value of ρ_min_, ρ_max_ and β_max_ in Equations (9), (11) and (12) is thus replaced with ρ_1min_, ρ_1max_ and β_1max_ for the determination of power of P_a1_ and these quantities are defined below: (25)$$\rho_{1min}=R\frac{{\sin\varepsilon }_{1}}{\cos\beta }$$

(26)$$\rho_{1max}=R\sin\theta_{1c}$$

(27)$$\beta_{1max}=\arccos\left(\frac{{\sin\varepsilon }_{1}}{sin\theta_{1c}}\right).$$

Substituting the integration limits from Equation (25) to (27) and applying these to the overlap region 1 (as represented in Figure 3b), the power density P_2_′ in Equation (13) is now written as Pa_1_′:(28)$$P_{a1}\prime =2\int_{\beta =0}^{\beta_{1max}}\int_{\rho_{1min}}^{\rho_{1max}}L\;co{s\theta }_{1}\;si{n\theta }_{1}\;{d\theta }_{1}\;{d\beta }_{1}.$$

Solving Equation (28) above yields the following result: (29)$$P_{a1}\prime =L\left(sin^{2}\theta_{1c}\beta_{1max\;}-\;sin^{2}\varepsilon_{1}tan\beta_{1max}\right).$$

Through applying similar approach, the power density within the overlapping region 2 is expressed as follows:(30)$$P_{a2}\prime =L\left(sin^{2}\theta_{2c}\beta_{2max\;}-\;sin^{2}\varepsilon_{2}tan\beta_{2max}\right).$$

Hence, the total power coupled into fiber 2 after summation of both integrals over the two different area element at the endface of the input fiber is thus rewritten as:(31)$$P_{2}=\pi a^{2}L\;\left(NA_{1}{}^{2}\beta_{1max\;}-\;sin^{2}\varepsilon_{1}tan\beta_{1max}\right)+\pi a^{2}L\;\left(NA_{2}{}^{2}\beta_{2max\;}-\;sin^{2}\varepsilon_{2}tan\beta_{2max}\right)\;.$$

Assuming the input fiber has NA_1_ and receiving fiber has NA_2_, the coupling ratio in Equation (21) is defined as:(32)$$\eta =\frac{1}{\pi NA_{1}{}^{2}}\;\left(NA_{1}{}^{2}\beta_{1max\;}-\;sin^{2}\varepsilon_{1}tan\beta_{1max}\right)+\frac{1}{\pi NA_{2}{}^{2}}\;\left(NA_{2}{}^{2}\beta_{2max\;}-\;sin^{2}\varepsilon_{2}tan\beta_{2max}\right).$$

## 4. Theoretical Calculation and Comparison with Experimental Data 

In this section, the concept of transverse offset and angular misalignment is applied to a fiber optic bending sensor based on an intensity interrogation, such as the one used in the investigation described in this article. The configuration of the sensor is presented in Section 4.1. The power coupling Equation (18) as expressed above is applied to provide the complete analytical solution to the light propagation in the sensor and the outcome is compared with the experimental data reported in Section 4.2 below. 

### 4.1. Sensor Configuration and Output Ratio Equation

The bending sensor described in this investigation is based on the concept of light transmission loss in angular misalignment from three output fibers that are transversely offset to align to the center axis of the input fiber. The tilt angle at the coupling region causes an alignment mismatch between the input and output fibers and hence the optical power is redistributed between the receiving fibers. The sensor utilized three identical step-index multimode fibers with 1 mm overall (core and cladding) diameter (core diameter = 980 µm) and aligned in the configuration as shown in Figure 4a. Using this configuration, it is possible to design an optical sensor for spine monitoring for measuring lumbar spine bending for two axis operation (lateral and sagittal). A transmitting fiber receives light from a red LED source of 660 nm center wavelength and is aligned in the center position of the three output fibers so that all output fibers ideally receive equal proportions of the transmitted optical power.

The transverse offset coordinates of the output fiber relative to the transmitting fiber as illustrated in Figure 4a are F1(−r, −r/2cos30°), F2(r, −r/2cos30°) and F3(0, r/cos30°), where F1, F2 and F3 refer to the three output fibers in Figure 4 and r is the radius of each identical fiber, which is 0.5mm in the case of the sensor of this investigation. 

For calculation of the light propagation, the three output fibers are subjected to an angular misalignment (tilt) of angle α in the lateral direction (x-axis) and the sagittal direction (y-axis). The ratio of the output light intensity from the three output fibers are calculated based on the coupling power received in each fiber. Using this intensity modulation method, the angular bending response in the lateral plane is calculated from the following output ratio equation *R(**α_x_)*: (33)$$R\left(\alpha_{x}\right)=\;\frac{F_{1}-F_{2}}{F_{1}+F_{2}}.$$

For the sagittal bending assessment (angular tilt in the y-axis), the angular bending response is calculated using the normalized output power ratio equation $R\left(\alpha_{y}\right)$: (34)$$R\left(\alpha_{y}\right)=\;\frac{\frac{1}{2}F_{1}+\frac{1}{2}F_{2}-F_{3}}{\frac{1}{2}F_{1}+\frac{1}{2}F_{2}+F_{3}}.$$

The normalized output ratio results for both lateral and sagittal bending transmission loss was calculated using Equations (33) and (34). The power transmission for each output fiber was further calculated using Equation (20). The output ratio equation for the lateral bending case in Equation (23) is therefore established as: (35)$$R\left(\alpha_{x}\right)=\frac{\left(NA^{2}\;\beta 1_{max}-sin^{2}\frac{\gamma_{1}}{2}\;tan\beta 1_{max}\right)-\left(NA^{2}\beta 2_{max\;}-sin^{2}\frac{\gamma_{2}}{2}tan\beta 2_{max}\right)}{\left(NA^{2}\beta 1_{max\;}-sin^{2}\frac{\gamma_{1}}{2}tan\beta 1_{max}\right)+\left(NA^{2}\beta 2_{max\;}-sin^{2}\frac{\gamma_{2}}{2}tan\beta 2_{max}\right)}.$$

And the sagittal bending ratio in Equation (24) can be represented as:(36)$$R\left(\alpha_{y}\right)=\frac{\left(NA^{2}\;\beta 1_{max}-sin^{2}\frac{\gamma_{1}}{2}\;tan\beta 1_{max}\right)+\left(NA^{2}\beta 2_{max\;}-sin^{2}\frac{\gamma_{2}}{2}tan\beta 2_{max}\right)-2(NA^{2}\beta 3_{max\;}-sin^{2}\frac{\gamma_{3}}{2}tan\beta 3_{max})}{\left(NA^{2}\beta 1_{max\;}-sin^{2}\frac{\gamma_{1}}{2}tan\beta 1_{max}\right)+\left(NA^{2}\beta 2_{max\;}-sin^{2}\frac{\gamma_{2}}{2}tan\beta 2_{max}\right)+2(NA^{2}\beta 3_{max\;}-sin^{2}\frac{\gamma_{3}}{2}tan\beta 3_{max})}$$
where the limiting values of *β* and *γ* in both equations are substituted based on the angular bending and axial misalignment offset of each output fiber. The ratio varies corresponding to the angular misalignment *α* in the x-axis (*α_x_* for lateral bending) and y-axis (*α_y_* for sagittal bending).

### 4.2. Experimental Test Results 

The comprehensive details of the sensor design and fabrication for spine monitoring are provided in a previous article by the authors of this article [22]. The sensor was mounted on a high precision rotational stage to allow actuation of accurate and repeatable angular bending. The sensor was tilted in both lateral and angular directions corresponding to the schematic diagram of Figure 4 for a wide range of bending angles, and the power coupling ratio of three output fibers corresponding to the values calculated in Equations (33) and (34) were captured experimentally during the measurements. Each test was repeated three times and the average output ratio value for each tilted angle was plotted and compared with the theoretical calculation results obtained from the equations of Section 4.1. 

## 5. Results and Discussion

The power coupling ratio versus angular misalignment angle response was evaluated and compared with the corresponding theoretical calculation results obtained from Equations (35) and (36). In the theoretical representation of the POF sensor in this investigation, the power transmission for each output fiber was calculated using Equation (20). The transverse offset of each output fiber relative to the input fiber was considered in the theoretical calculation by substituting the coordinates of each output fiber, as illustrated in Figure 4. The power ratio was plotted over the entire angular bending angle for lateral plane (x-axis) and sagittal plane (y-axis). Figure 5 shows a comparison of the theoretical calculations and experimental results for angular bending in the lateral and sagittal planes. 

For the theoretical analysis, the output ratio was calculated using the numerical aperture (NA) of the step-index multimode fiber with the value of 0.51 supplied as per the manufacturer’s specification data. Figure 5 shows the trend of both bending measurements and both cases exhibit excellent agreement when compared with the theoretical values if modeled with a separation gap of 1.34 mm between the input and output fibers. 

Table 1 summarizes the maximum and minimum deviation between the theoretical and experimental results across the working range plotted in Figure 5. For lateral bending, the maximum deviation value was 0.0877 and occurs at the bending angle of −2°. The minimum deviation between both results occurs at 9° with the output ratio difference being as small as 0.0037 (absolute value). In the case of sagittal bending, the maximum deviation between the theoretical and experimental results is at the output ratio value of 0.12 that occurs at 9° of flexion. The minimum deviation between both results falls at 4° with an output ratio difference of 0.006 (absolute value). If the output ratio is offset to the range from zero to two, any percentage deviation between the theoretical and experimental results is calculated using the following equation: (37)$$Percentage\;Deviation\;\left(\%\right)=\frac{Ratio\left(theory\right)-Ratio\left(\exp\right)}{Ratio\;\left(theory\right)}\;x\;100\%$$

The maximum deviation between the theoretical and experimental results are calculated to be 11.97% difference (at a −2° tilt angle) for the lateral plane and 6.04% (at 9° tilt angle) for the sagittal plane. The slight deviation between both results may be caused by human error during manual fabrication of the sensor.

In the experiment, the designated gap between the input and output fibers was set to be around 1.10 mm to allow some space for fiber bending. However, the separation (gap) between both tubes might extend a further small amount during the alignment process between both fiber tubes and during the bending. The useful operating angular range is between ±12° for both lateral and sagittal bending in the case of the sensor of this investigation, which was determined experimentally in reference [22]. The bending angle beyond this working limit results in saturation of the sensor output. 

Based on the theoretical calculation, the sensitivity and operating range of the sensor can be adjusted according to the inter-fiber separation gap (between input and output fibers) for other application purposes. Figure 6 shows the result of theoretical calculation of varying the separation gap effect on the bending angle response. The output ratio was plotted against the bending angle for four different normalized gap ratios. The value of the normalized gap ratio, ε was calculated from the separation gap divided by the value 1.34 mm, which is the theoretical estimated gap that fits the response of the sensor fabricated in this investigation. For angular bending in both directions, a larger separation gap (higher normalized gap ratio) results in a larger sensing range. However, the sensitivity of the sensor is reduced as a compromise for a larger working range. For example, in the lateral bending ratio, the normalized separation gap ε of 1.5 constitutes a maximum operating range of ±29° but with a lower sensitivity response of 0.0466/1°. On the other hand, the output ratio of the sensor with normalized separation gap ε of 0.9 has a lower working range of ±5°, but with a higher sensitivity response of 0.2767/1°. From these theoretical calculations, the bending response of the sensor can be adjusted in the future depending on the desired operating range or sensitivity for a particular application. 

Figure 7 demonstrates the response of the sensor using different NA of the multimode fibers. In order to calculate the response of the sensor’s output based on different NA of the step-index fiber used, other correlated parameters were assumed constant. The gap between both input and output fibers was fixed at 1.34 mm, which is the theoretical estimated gap that fits the response of the sensor fabricated in this investigation. The result shows that using a fiber with higher NA results in a higher working range (up to around ±30° for fibers with NA of 0.65) as the receiving fibers are still able to receive light from the input fiber even when tilted to a relatively high angle. However, the sensitivity is greatly reduced in this case. On the other hand, using a fiber optic with low NA will limit the operation range for the sensor, but provide a higher sensitivity. 

Another aspect that can be considered in choosing the suitable parameters for the sensor is the case in which both input and output step-index multimode fibers have a different NA. For this analysis, the power coupling equation has been derived in Section 3.1 and Equation (31) was used to calculate the total power coupled into each of the output fibers. Using a similar approach, Equations (33) and (34) were used to calculate the output ratio of a sensor for the two different bending axis. Figure 8 illustrates the response of the sensor when the input fiber and output fibers have a different NA. NA1 represents the numerical aperture of the input fiber while NA2 remained fixed throughout the calculation study, which was 0.51 in this case. The sensor demonstrates a smaller operating range with a higher sensitivity when NA1 is small. 

When the results of Figure 8 are compared with Figure 7, the case where NA1 < NA2 with NA1 = 0.45 and NA2 = 0.51 (blue line in Figure 8) has a higher operating range of up to ±8° compared to the case when all fibers have NA of 0.45 (blue line in Figure 7), which has a much smaller operating range of around ±3°. However, this value is smaller than the case where all fibers have the same NA of 0.51 (red dotted curve in Figure 7) for which the operating range is ±10°. For the case where NA1 > NA2 (where NA1 is 0.60 and NA2 is 0.51), the sensor response (sensitivity and operating region) lie in between the case where all fibers have NA of 0.51 and 0.60. This allows the creation of more flexible options in selecting the right parameters for a particular application.

Using the theoretical model developed in this paper, these parameters can be used to more effectively and rapidly design the sensor for a given set of requirements. If the bending angle exceeds the working limit (high angular misalignment), this results in a saturation of the sensor output where the output fiber receives a negligible amount of light coupled from the input fiber. The theoretical study is therefore important for matching the sensor to the desired characteristic, such as the bending angle range or sensitivity, depending on the application.

## 6. Conclusion

The power coupling ratio between the step-index multimode fibers for transverse and angular misalignment in an optical fiber bend sensor has been calculated from a purely analytical approach. Theoretical calculation was applied to a fiber sensor with a purely transverse offset as well as where the input fiber is displaced at an angle relative to the output fibers in the lateral (x-axis) and sagittal (y-axis) directions. The theoretical calculations show excellent agreement with the experimental results across the working (angle) range of the sensor. The percentage difference between the theoretical and experimental results calculated in the worst case scenario is 11.97% at a −2° tilting angle for the lateral plane (x-axis) and 6.04% at a 9° tilting angle for the sagittal plane (y-axis). The effect of different parameters on the sensor performance, including the fiber gap and numerical aperture, has been demonstrated. The establishment of the theoretical analysis described in this article is a significant development for choosing the parameters for a sensor that operates based on purely transverse displacement, angular misalignment, or both, as well as for the case in which both fibers have a different NA. The analysis is also applicable for calculating power loss in optical fiber connectors or splices for both transverse and angular misalignment scenarios. 

## Figures and Tables

**Figure 1 sensors-19-04968-f001:**
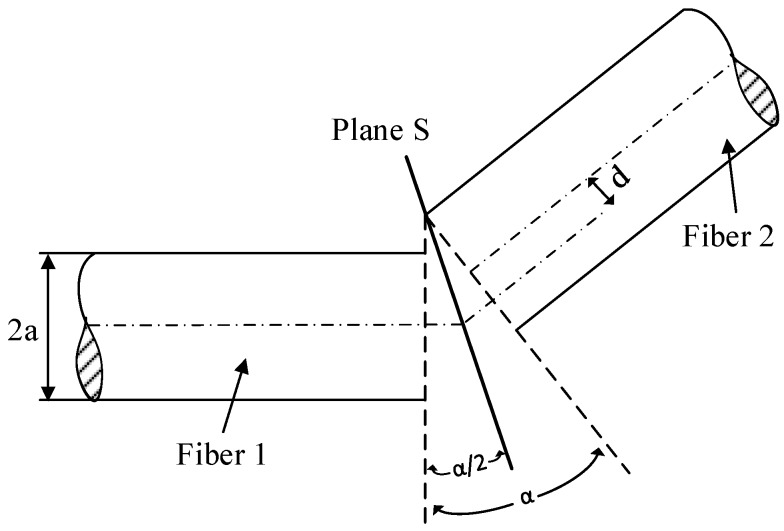
Graphical definition of lateral offset and angular misalignment between two fibers.

**Figure 2 sensors-19-04968-f002:**
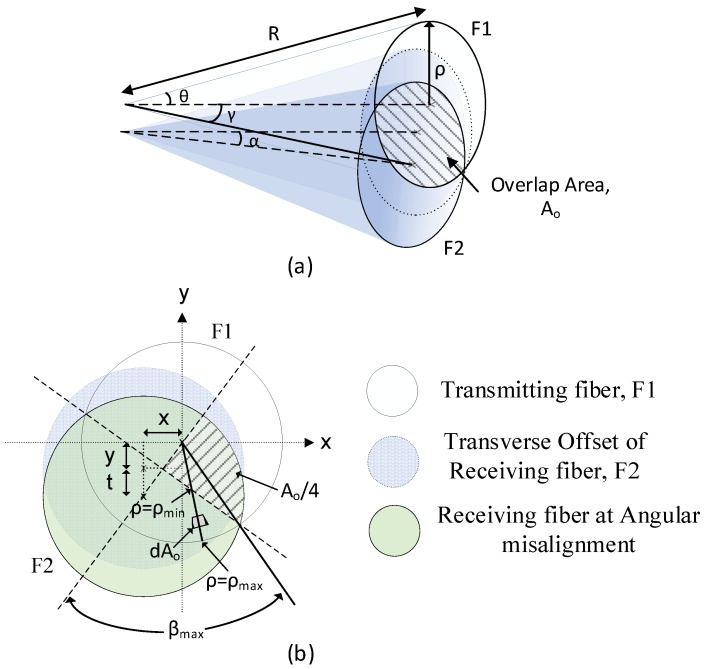
Overlapping cones of numerical aperture in (**a**) side view and (**b**) plan view.

**Figure 3 sensors-19-04968-f003:**
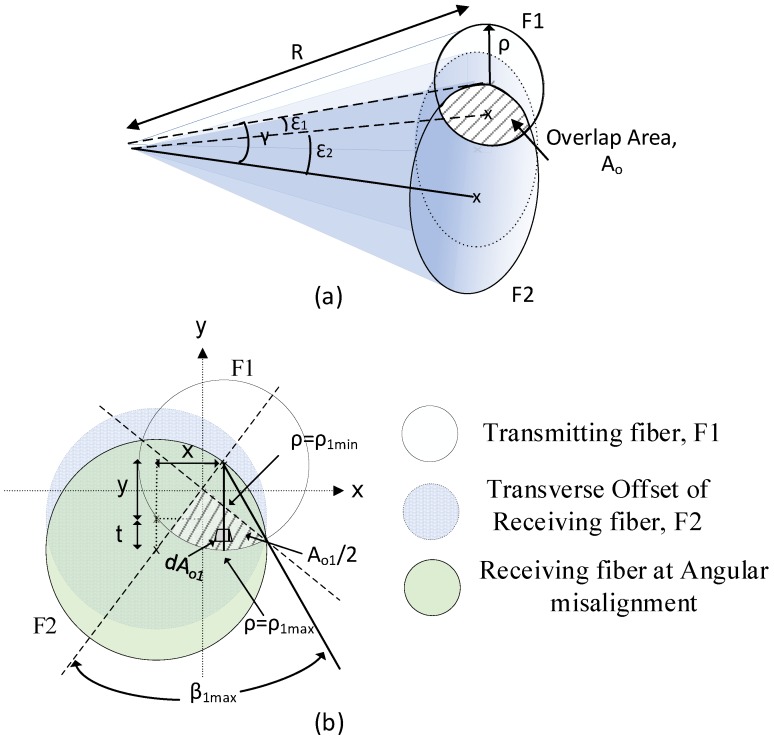
Overlapping cones of different numerical aperture between input and output fibers in (**a**) side view and (**b**) plan view.

**Figure 4 sensors-19-04968-f004:**
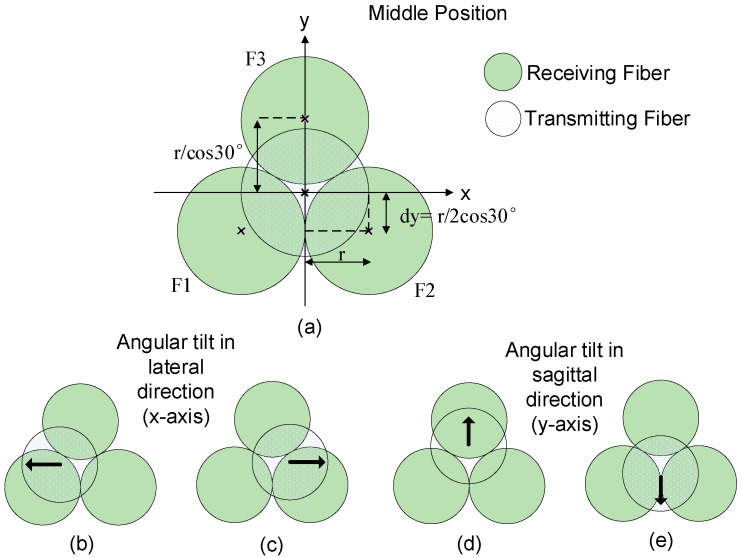
Position of transmitting fiber and three receiving fiber at different lateral offset in (**a**); subjected to angular bending (misalignment) in the x-axis or lateral plane in (**b**) and (**c**); angular bending in the y-axis or sagittal plane directions in (**d**) and (**e**).

**Figure 5 sensors-19-04968-f005:**
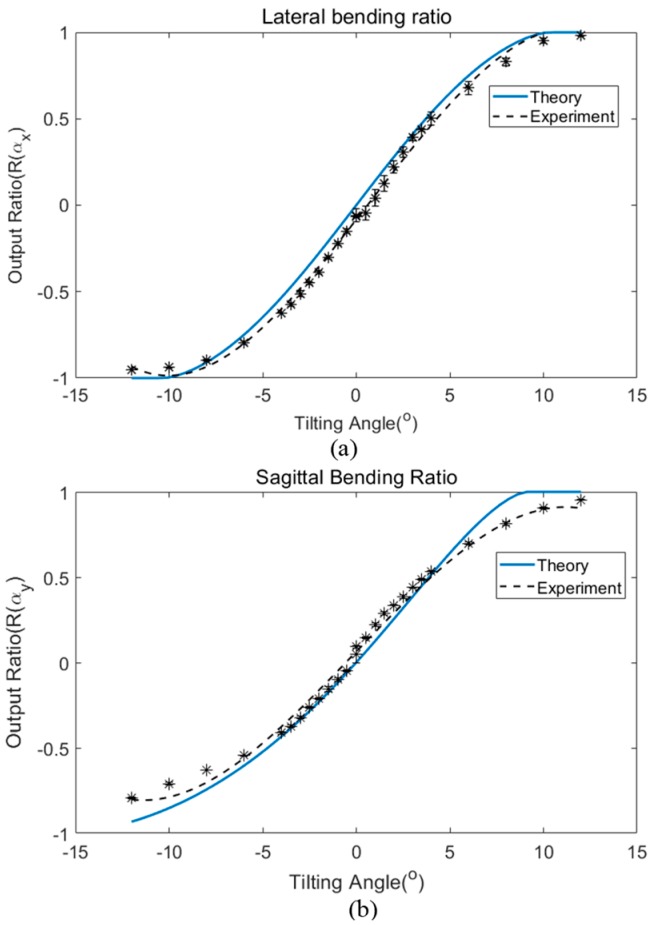
Comparison of the power coupling ratio of the three laterally misaligned output fibers between the theoretical and experimental results for angular tilt in (**a**) lateral and (**b**) sagittal planes.

**Figure 6 sensors-19-04968-f006:**
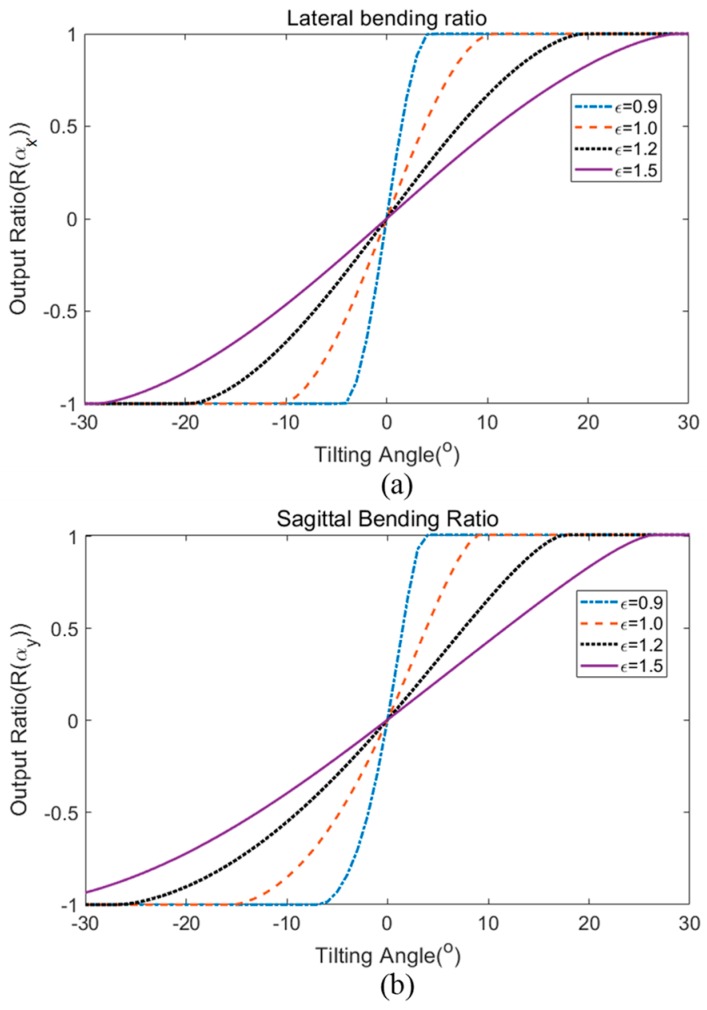
Theoretical calculation of the effect of the inter-fiber gap on the output ratio response for (**a**) lateral bending and (**b**) sagittal bending.

**Figure 7 sensors-19-04968-f007:**
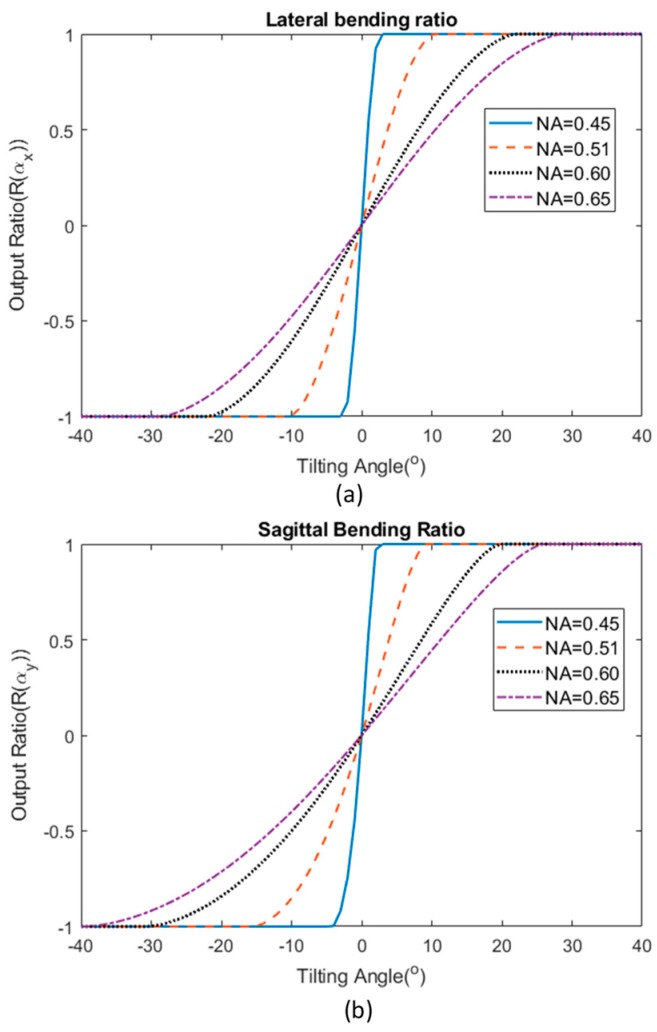
Theoretical estimation of the sensor’s bending response when using fiber optic with different numerical aperture (NA) for (**a**) lateral bending and (**b**) sagittal bending.

**Figure 8 sensors-19-04968-f008:**
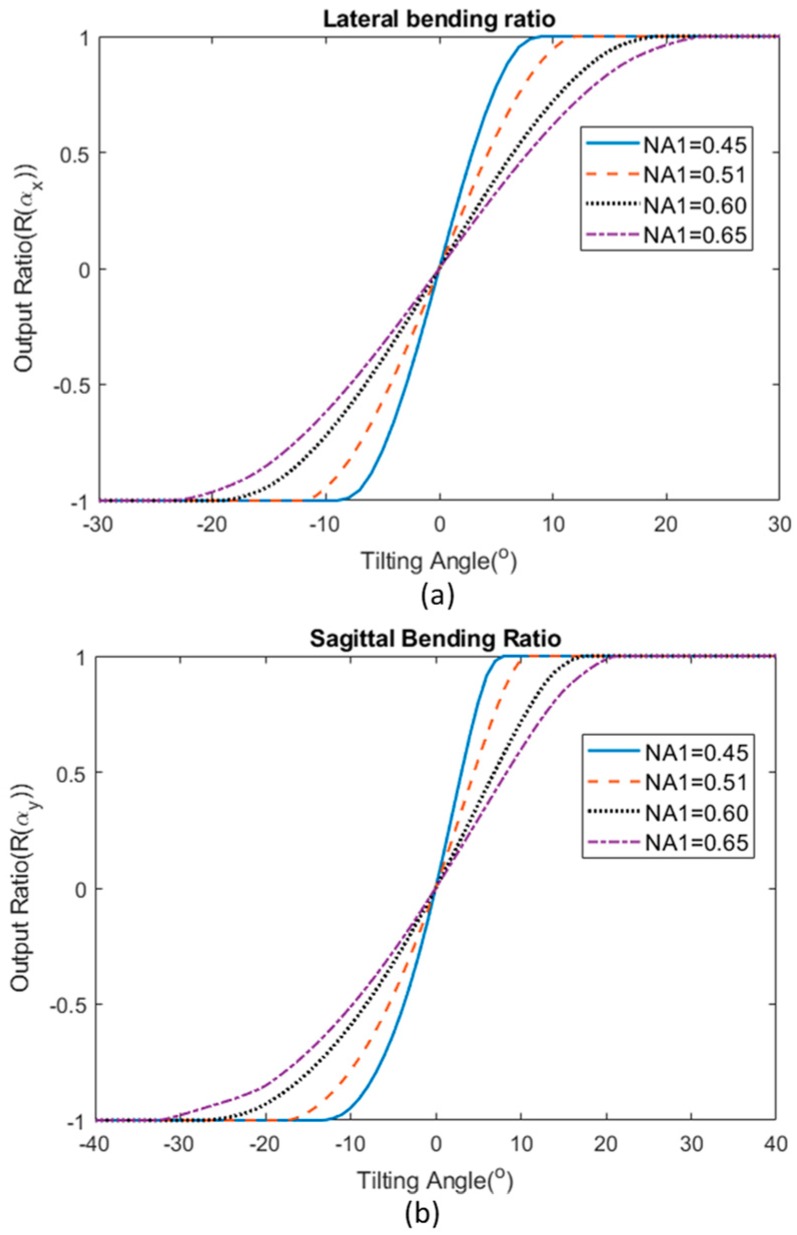
The sensor’s bending response of (**a**) lateral bending and (**b**) sagittal bending when the input fiber and output fibers have a different NA, where NA1 is the numerical aperture of the input fiber and NA2 is the NA of output fibers, which is 0.51 in this case.

**Table 1 sensors-19-04968-t001:** Maximum and minimum percentage difference between the theoretical and experimental results across the working range (±12°).

	Maximum Percentage Difference			Minimum Percentage Difference		
	Bending Angle (°)	Output Ratio Difference	Percentage Deviation	Bending Angle (°)	Output Ratio Difference	Percentage Deviation
Lateral Bending (X-axis)	−2 *	0.087704	11.9%	9	0.0037607	0.193%
Sagittal Bending (Y-axis)	9	0.119962	6.04%	4	0.006099	0.408%

* the value at a bending angle of 0° could not be used as the output is set to read 0 at this point.

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
