# Peer review of "An Analytical Model for Describing the Power Coupling Ratio between Multimode Fibers with Transverse Displacement and Angular Misalignment in an Optical Fiber Bend Sensor"

_sensors, 2019, doi:10.3390/s19224968_

Round 1

Reviewer 1 Report

Comments about the article

It is not clear how Fig. 1 is related to Fig. 2 In the theoretical section, you used just two fiber; however, in the development of the sensor, you used four fibers, why this discrepancy?

Author Response

Response to Reviewer 1 Comments

Comments 1.  It is not clear how Fig. 1 is related to Fig. 2 In the theoretical section, you used just two fiber; however, in the development of the sensor, you used four fibers, why this discrepancy?

Response 1: We apologize for the unclear description relating to the two figures in question. Fig. 1 shows the general graphic of two fibers with lateral offset and angular misalignment.  Fig. 2 focuses on the view of two extended numerical apertures from both input and output fibers including the required parameters for the theoretical calculation when the fibres undergo misalignment. In the theoretical section, two fibers are used for a simplified explanation and derivation of the loss resulting from two misaligned fibers. In the case of the real sensor, we used four fibers (three output and one input) in order to fabricate a spine monitoring sensor that allows simultaneous bending measurement in two axis, x- and y-axis (lateral and sagittal). An explanatory comment has been added into the paper in section 4.1 paragraph (line 205-207): ‘Using this configuration ……’.

Reviewer 2 Report

The theoretical calculation of the power coupling ratio between the step-index multimode fibers for transverse and angular misalignment in an optical fiber bend sensor has been provided in this paper. However, for most of the sensors, the accurate analytical calculation of the transmission power (loss) is not an critial for improving their performance. The motivation of the study is not clarified. The author should specified how the sensitivity or the measurement range could be improved by using the theoritical calculation.

Author Response

Response to Reviewer 2 Comments

Comments 1.  The theoretical calculation of the power coupling ratio between the step-index multimode fibers for transverse and angular misalignment in an optical fiber bend sensor has been provided in this paper. However, for most of the sensors, the accurate analytical calculation of the transmission power (loss) is not an critial for improving their performance. The motivation of the study is not clarified. The author should specified how the sensitivity or the measurement range could be improved by using the theoritical calculation. 

Response 1: We apologize for the unclear description on the motivation of this study, and thank you for pointing out the problem. To further strengthen the motivation of the study, we have specified how the sensitivity or the measurement range could be improved using the theoretical calculation as suggested by the reviewer. We have also added extra analysis results in order to specify how the sensitivity and operating range can be chosen or optimized based on required application. These include the variation of the inter-fiber gap in the sensor configuration, the choice of numerical aperture (NA) of the fibers including the case in which both step-index multimode fibers have different NA. A new section 3.1 has been included in the manuscript to describe and derive the equation of power coupling lost in the case where both multimode fibers have different NA values. These have been added in the article in section 5 paragraph 6, (commencing at line 295): ‘Based on the theoretical estimation ……’. The new analysis results have been added as new Figure 6, 7 and 8 in the article.

Reviewer 3 Report

The authors propose an analytical model to describe the power-coupling ratio between multimode fibers with transverse displacement. They include a mathematical approach to explain the coupling phenomena and obtain an analytical expression as function of simple parameters. In addition, I section 5, they compare the obtained results with the results obtained in a previous experiment. However, I cannot identify the main contribution of this study compared with others. For the above, I think that the manuscript in the current form is not suitable to be published in sensors journal. I suggest the following changes in the manuscript:

The constitution of the proposed paper is not clear. Actually, I think that is not novelty since other work reported a similar study in the past. Thus, Can you explain the main contribution of your work? I believe that it is not clear. For example, can you detail the main differences of your model compared with the models of ref. 14 and ref. 15. In addition, can you explain the difference of the proposed study compared with the following papers: DOI:10.1364/OE.23.022318 DOI: 10.1364/AO.15.002765 DOI: 10.1109/JLT.2013.2273492 DOI:10.1364/OE.23.008061 DOI: 10.3390/photonics6030088 The author made some important assumptions in order to simplify the model. One of them was the following: “all fibers in the calculation & experiment are identical step-index multimode fibers with the same numerical aperture (NA) and core diameter”. However, I think that your work could be more interesting if you complete the analysis for the case in which both step-index multimode fibers have different NA (For Example in ref. 15, Gao et.al. make an analysis in this situation). It could have a strong impact in optical fiber sensor research since explain the most common situation. Thus, I suggest the inclusion of this case in the manuscript.

In addition, you can include other two cases in the manuscripts. The first one when NA1>NA2 and the other one when NA1<NA2.

In section 5, the authors compare the results obtained with the proposed analytical model with experimental results. A more interesting alternative is the inclusion of a curve of results obtained with a numerical approach. Can you include another experimental verification of your analytical model?

Author Response

Response to Reviewer 3 Comments

General Comment: The authors propose an analytical model to describe the power-coupling ratio between multimode fibers with transverse displacement. They include a mathematical approach to explain the coupling phenomena and obtain an analytical expression as function of simple parameters. In addition, I section 5, they compare the obtained results with the results obtained in a previous experiment. However, I cannot identify the main contribution of this study compared with others. For the above, I think that the manuscript in the current form is not suitable to be published in sensors journal. I suggest the following changes in the manuscript:

The constitution of the proposed paper is not clear. Actually, I think that is not novelty since other work reported a similar study in the past. Thus, Can you explain the main contribution of your work? I believe that it is not clear. For example, can you detail the main differences of your model compared with the models of ref. 14 and ref. 15. In addition, can you explain the difference of the proposed study compared with the following papers: DOI:10.1364/OE.23.022318 DOI: 10.1364/AO.15.002765 DOI: 10.1109/JLT.2013.2273492 DOI:10.1364/OE.23.008061 DOI: 10.3390/photonics6030088 The author made some important assumptions in order to simplify the model. One of them was the following: “all fibers in the calculation & experiment are identical step-index multimode fibers with the same numerical aperture (NA) and core diameter”. However, I think that your work could be more interesting if you complete the analysis for the case in which both step-index multimode fibers have different NA (For Example in ref. 15, Gao et.al. make an analysis in this situation). It could have a strong impact in optical fiber sensor research since explain the most common situation. Thus, I suggest the inclusion of this case in the manuscript.

In addition, you can include other two cases in the manuscripts. The first one when NA1>NA2 and the other one when NA1<NA2.

In section 5, the authors compare the results obtained with the proposed analytical model with experimental results. A more interesting alternative is the inclusion of a curve of results obtained with a numerical approach. Can you include another experimental verification of your analytical model?

Response:  We apologize for not being able to provide clear constitution of the proposed article. The main motivation of the work is to use the simple analysis of power coupling ratio along with the fiber configuration to optimized or design a sensor using the derived equation. We are applying the concept of power loss into a low cost spine bending monitoring sensor suitable for clinical use, based on simple intensity interrogation. The difference of the proposed study with the listed papers is the application of power coupling concept into a bending monitoring sensor.

We fully agree with this comment and following your recommendations we have added another section of analysis for the case in which both step-index multimode fibers have different NA values. This has been added as a new subsection 3.1 “Solution for two step-index multimode fibers of different numerical aperture”.

In order to strengthen the motivation of this study, we have added further analysis on the sensor’s output response when the configuration is subject to a variable distance (gap) between the input and output fibers, as well as the case  of fibers with different NA. In addition, we have followed the reviewer’s suggestion on adding additional results demonstrating fiber’s response when the input and output of sensor have different NA. This has been added in in the results and discussion section 5 paragraph 6, starting (commencing line 295): ‘Based on the theoretical estimation ……’. The new analysis results have been added as Figure 6, 7 and 8 in the article.

Reviewer 4 Report

The authors showed an analytical approach using step-index multimode fibers and discussed its parameters impact on its sensing performance for bending. In my opinion, it contributes little to the optical fiber sensor fields. Therefore, I suggest rejection for its publication on SENSORS.
Some comments are listed below:

1. In the introduction section, the authors should review more works on comparing and discussing the optical fiber bend sensor, POF sensor, and the fiber misalignment. In my opinion, the power coupling theories between the two fibers have been researched in the past decades.

2. Although the authors raised their previous work, I can not find what can be improved with the results shown in this report.
This paper seems not to have any novel information, particularly.

3. The author compared theoretical results and experimental results. However, there are no experimental setup and conditions. I can not understand how the authors controlled the tilt angle of the fiber.

4. Why did the authors use the three fiber for receiving transmission light?

5. I suggest the authors demonstrate the more experiment for bend fiber rather than numerical simulation. The experimental result (data: only three times) is too little (only figure 4, and table 1). This amount of results corresponds to conference proceedings at most.

Author Response

Response to Reviewer 4 Comments

The authors showed an analytical approach using step-index multimode fibers and discussed its parameters impact on its sensing performance for bending. In my opinion, it contributes little to the optical fiber sensor fields. Therefore, I suggest rejection for its publication on SENSORS.

Some comments are listed below:

Comments 1. In the introduction section, the authors should review more works on comparing and discussing the optical fiber bend sensor, POF sensor, and the fiber misalignment. In my opinion, the power coupling theories between the two fibers have been researched in the past decades.

Response 1:  We apologize for the unclear description on the motivation of this study and we appreciate the feedback from the reviewer. Our motivation in this study is to focus more on using a simplified power coupling theory and apply this to a real life spine bending monitoring sensor based on the previously reported experimental sensor configurations. We agreed with the reviewer’s comment and as suggested by the reviewer, a further review on existing literature has been included discussing optical fiber bend sensors  and POF as an extra paragraph in the introduction section 1, paragraph 3 (commencing line 45): ‘For the case of  fiber……’. And the following new references have been included

14.   Gouveia, C.; Jorge, P.; Baptista, J.; Frazao, O., Temperature-independent curvature sensor using FBG cladding modes based on a core misaligned splice. IEEE Photonics Technology Letters 2011, 23, (12), 804-806.

15.   Kong, J.; Ouyang, X.; Zhou, A.; Yu, H.; Yuan, L., Pure directional bending measurement with a fiber Bragg grating at the connection joint of eccentric-core and single-mode fibers. Journal of Lightwave Technology 2016, 34, (14), 3288-3292.

16.   Dunne, L. E.; Walsh, P.; Hermann, S.; Smyth, B.; Caulfield, B., Wearable monitoring of seated spinal posture. IEEE transactions on biomedical circuits and systems 2008, 2, (2), 97-105.

17.   Bilro, L.; Oliveira, J.; Pinto, J.; Nogueira, R., A reliable low-cost wireless and wearable gait monitoring system based on a plastic optical fibre sensor. Measurement Science and Technology 2011, 22, (4), 045801.

18.   Zawawi, M.; O'Keeffe, S.; Lewis, E., Plastic optical fibre sensor for spine bending monitoring with power fluctuation compensation. Sensors 2013, 13, (11), 14466-14483.

Comments 2.  Although the authors raised their previous work, I can not find what can be improved with the results shown in this report.

This paper seems not to have any novel information, particularly.

Response 2:  We apologize for not being able to clarify the novelty of the work. To strengthen the motivation of this study, we have added extra analysis and results on the simulation side to show how the analysis results aid in designing a sensor for different applications. This includes adding additional analysis of the sensor’s response (sensitivity and operating range) based on the variation of the inter-fiber gap in the sensor configuration, the chosen numerical aperture (NA) of the fiber and the case in which both step-index multimode fibers have different NA values. A new section 3.1 has also been included in the manuscript to describe and derive the equation of power coupling lost in the case where both multimode fibers has different numerical aperture. These have been added in the article in section 5 paragraph 6, (commencing at line 295): ‘Based on the theoretical estimation ……’. The new analysis results have also been added as Figure 6, 7 and 8 in the article.

In the paper we demonstrate the ability for the sensor configuration to work as a bending monitoring sensor using the theoretical calculation. Based on the simulation study, there are more future applications that can be accessed through selecting appropriate parameters e.g. For knee bending measurements which require a larger operating bending range by designing a larger gap between the input and output fibers, or a smaller gap for better sensitivity for breathing measurements.

Comments 3.  The author compared theoretical results and experimental results. However, there are no experimental setup and conditions. I can not understand how the authors controlled the tilt angle of the fiber.

Response 3:  We apologize for not being able to list out the details of the experiment as the details of the sensor fabrication for spine monitoring are provided in a previous article by the authors of this article entitled ‘Low cost portable 3-D printed optical fiber sensor for real-time monitoring of lower back bending’[22]. This has been mentioned in section 4.2, paragraph 1, line 1 (commencing on line 240):‘ The comprehensive details ……’.

In the experiment, the sensor was placed on an optical setup that consists of translational stage and a precise rotational stage for step size of 0.5o as illustrated in the figure below. The signal obtained from the sensor was recorded throughout the measurement with the stage rotated at the step interval of 0.5o.

"Please see the attachment for the figure."

Ref:

[22]        Kam, W.; O'Sullivan, K.; O'Keeffe, M.; O'Keeffe, S.; Mohammed, W. S.; Lewis, E., Low cost portable 3-D printed optical fiber sensor for real-time monitoring of lower back bending. Sensors and Actuators A: Physical 2017, 265, 193-201.

Comments 4. Why did the authors use the three fiber for receiving transmission light?

Response 4:  For the development of the sensor, we  used four fibers (three outputs and one input) in order to fabricate a spine monitoring sensor that allows simultaneous bending measurement in two axes, x-axis and y-axis (lateral and sagittal). These have been added into the paper in section 4.1 paragraph 1 line 6 (commencing at line 205): ‘Using this configuration ……’. Since all the input and output fibres in the calculation and experiment are identical, it can be shown from a theoretical calculation that both of the power ratio equations are independent of the light source intensity and loss (optical attenuation) of the system. Therefore, this configuration (3 output and 1 input fiber) allows the sensor to greatly eliminate noise from the light source and environment using the output ratio equations (33) and (34) in the article.

Comments 5. I suggest the authors demonstrate the more experiment for bend fiber rather than numerical simulation. The experimental result (data: only three times) is too little (only figure 4, and table 1). This amount of results corresponds to conference proceedings at most.

Response 5: Thank you for the suggestion. However the reason that we are not focusing on the experimental part is as the paper tend to focusses on the theoretical analysis data that when applied in the sensor configurations, thus opening the possibility for optimization of the sensor for future applications. Full experimental results can be found in ref [22], where the sensor was 3-D printed using Stratasys Connex 500 printer which can be performed repeatedly. The fabricated unit is manufactured to a high resolution and hence able to accurately and repeatedly align the fibers in the unit and further to hold them still during measurement.

Reviewer 5 Report

The authors present a theoretical study on the alignment between two fibers and compare the medellum with the experimental results using a previously published setup (reference 17). Since this article is a theoretical part of the coupling study and its impact on the final results, it should be clear that the study focuses on the application of the reference article 17. Throughout the manuscript, the authors do not mention what is the advantage of the presented model and what solution to solve the alignment problems.

This study, is for the tested sensors in reference 17? What is the impact on reducing the number of fibers and the diameter? And, if change the applications, the results are the same? The authors refers in conclusion the possibility of this analysis can be used for calculating power loss in optical fiber connectors or splices, but in manuscript don’t show.

The reference list needs to be improve. More reference about alignment is required. The authors refer other study but not compare the results. In mauncript is present several equations, but the authors not introduce reference. These equations were obtained by authores?

I reocommend the authors rewrite the manuscript and present a new version whhere need clarify the previous questions. I recommend a major review.

Author Response

Response to Reviewer 5 Comments

General Comment 1: The authors present a theoretical study on the alignment between two fibers and compare the medellum with the experimental results using a previously published setup (reference 17). Since this article is a theoretical part of the coupling study and its impact on the final results, it should be clear that the study focuses on the application of the reference article 17. Throughout the manuscript, the authors do not mention what is the advantage of the presented model and what solution to solve the alignment problems.

Response 1:

As all the input and output fibres in the calculation and experiment are identical, it can be shown from a theoretical calculation that both of the power ratio equations are independent of the light source intensity and loss (optical attenuation) of the system. If the type of fibre is fixed with all outputs received from the same source, the sensitivity of the output ratio is only dependent on lateral and angular offset between the input and output fibres. Using the ratio equation to estimate the bending angle, no reference signal is required as all the output fibres automatically provide a self-referenced signal. Noise from the light source or environment is therefore greatly reduced or eliminated considering that all three output fibres are closely bundled together and the input light signal is coupled from a single source. Moreover, the overall size of the sensor can be reduced as the requirement for a separate reference fibre in the sensor has been eliminated.

General Comment 2: This study, is for the tested sensors in reference 17? What is the impact on reducing the number of fibers and the diameter? And, if change the applications, the results are the same? The authors refers in conclusion the possibility of this analysis can be used for calculating power loss in optical fiber connectors or splices, but in manuscript don’t show.

Response 2: We apologize for the unclear description on the motivation of study, and thank you for pointing out the problem. With regards to the first question, the study aims to focus on the theoretical analysis, when applied into the configurations, opening the possibility for the sensor to be used for different future applications based on specific requirements (e.g. operating range and sensitivity).

Regarding the second question, the purpose of using three output fibers and one input fiber is to allow the measurement of bending angle in both axis direction, x-axis and y-axis (lateral and sagittal). Reducing the number of fibers (down to 2 output fibres) would allow the sensor to be able to measure bending in only one axis direction. Reducing the diameter of the output fiber would reduce the operating range which would render it no longer suitable for spine monitoring (measurement of bending angle in ±12º in our case). To further explain the impact of the sensor when using different parameters of fiber, we have added extra analysis results for aiding in designing sensors for different applications. This includes adding another analysis of the sensor’s response (sensitivity and operating range) based on variation of the inter-fiber gap in the sensor configuration, the chosen numerical aperture (NA) of the fiber and in the case in which both step-index multimode fibers have different NA. A new section 3.1 has also been included in the manuscript to describe and derive the equation of power coupling lost in the case where both multimode fibers has different numerical aperture. These has been added in the article in section 5 paragraph 6, starting (commencing on line 295): ‘Based on the theoretical estimation ……’. The new analysis results have also been added as Figure 6, 7 and 8 in the article.

In this article, more focus has been placed on a bending fiber sensor instead of fiber connectors or splices, and therefore the use of the theoretical analysis in those application will be included in future work. 

General Comment 3: The reference list needs to be improve. More reference about alignment is required. The authors refer other study but not compare the results. In mauncript is present several equations, but the authors not introduce reference. These equations were obtained by authores?

I reocommend the authors rewrite the manuscript and present a new version where need clarify the previous questions. I recommend a major review.

Response 3:  Thanks for the comments. We have added a new paragraph to discuss more about fiber optic sensor in the introduction section 1 paragraph 2 and thus new references from [14] to [18] have also been added in the article. The reason we are not comparing with previous results is that the sensor of this investigation is unique in terms of its compactness, operation and design.  A comparison for clinical measurement was made using a commercial clinical goniometer as discussed in ref 21, but even this comparison was limited due to the completely different sizes of the instruments, the goniometer be in much large and hence accessing different lumbar locations.  Also the equations referred to by the reviewer were developed for this sensor by the authors.

Round 2

Reviewer 3 Report

After the revision process, the manuscript is well written and well presented. The results are satisfactory and the quality was improved based on the comments of all reviewers. Therefore I am recommending it to be published unaltered.

Author Response

We appreciate the positive feedback from the reviewer. Thank you.

Reviewer 5 Report

The authors present a new version of the manuscript where clarify several questions presented in the previous report

. I suggest the authors introduce references in the equations.

Author Response

Thank you for the suggestion. For the response to the reviewer, we have added a sentence in section 3, paragraph 1 (commencing line 96) :"Based on the principles of ..." to clarify the issue. A reference [19] has also been included for equation 1 (line 99) in the manuscript as suggested by the reviewer.